# Chestnut Starch Nanocrystal Combined with Macadamia Protein Isolate to Stabilize Pickering Emulsions with Different Oils

**DOI:** 10.3390/foods11213320

**Published:** 2022-10-23

**Authors:** Jingyi Zheng, Lei Zhao, Junjie Yi, Linyan Zhou, Shengbao Cai

**Affiliations:** 1Faculty of Food Science and Engineering, Kunming University of Science and Technology, Kunming 650500, China; 2Beijing Engineering and Technology Research Center of Food Additives, Beijing Technology and Business University, Beijing 100048, China

**Keywords:** chestnut starch nanocrystal, macadamia protein isolate, Pickering emulsion, emulsion stability, effective delivery system

## Abstract

This study investigated the formation and molecular interaction mechanism of chestnut starch nanocrystal (SNC)/macadamia protein isolate (MPI) complexes and their application in edible oil-in-water Pickering emulsion (PE). SNC/MPI complexes were characterized by scanning electron microscopy and particle size analyzer. The PEs stabilized by SNC/MPI complexes were characterized by confocal laser scanning microscopy and rheological measurement. The results showed that hydrogen bonds between the two particles significantly affected the secondary structure and assembly of SNC/MPI complexes at the oil/water interface. The optimal mass ratio of SNC to MPI in the complexes with the best stability was determined as 20:1. The formation of edible oil-in-water PEs stabilized by SNC/MPI complexes significantly improved the oxidative and storage stability of different edible oils (olive oil, walnut oil, edible tea oil, and macadamia oil). These different edible oil-in-water PEs stabilized by SNC/MPI could be used as effective carriers of quercetin with their loading rates higher than 93%.

## 1. Introduction

Pickering emulsion (PE) stabilized by solid particles of colloidal size can act as an important delivery system for bioactive compounds [1]. Compared with traditional emulsions, the preparation of PEs avoids the use of surfactant, making the prepared emulsions more environmentally friendly and safe [2]. Therefore, the development of natural, biodegradable, safe, healthy, and food grade PE stabilizers is a major topic and great challenge in the field of food and pharmaceutical industries. Organic particles composed of carbohydrates and protein have attracted increasing interest as an alternative to synthetic particle stabilizers [3].

Starch is usually modified by chemical methods such as acid treatment, esterification, oxidation, and cross-linking [4]. Starch nanocrystals (SNCs) obtained from sulfuric acid hydrolysis of starch have many advantages including that they are inexpensive, abundant, biocompatible, biodegradable, non-toxic [5], and are deemed a promising food grade PE granule stabilizers for oil-in-water emulsions [6]. Compared with original starch, SNCs prepared from acid-hydrolysis of starch have higher zeta potential and better dispersion properties [7]. However, the presence of numerous hydroxyl groups in SNC can induce supramolecular-interaction-mediated aggregation of starch [8]. Macadamia protein isolate (MPI) is a promising natural emulsifier because of its amphiphilic properties and potential application in vegetarian and vegan products. However, emulsions stabilized by MPI alone are unstable because of their low water solubility and tendency to aggregate [9]. It has been shown that the use of protein-starch complexes rather than protein or starch alone improves the stability of emulsions, as well as the oxidative stability and bioavailability of encapsulated bioactive substances [10,11]. Compared with protein-stabilized emulsion, the addition of long-chain starch provided a more steric hindrance for stabilizing emulsion and changed the surface wettability of protein, so that the complex could better adsorb on the oil-water surface and form a dense interface film to prevent oil droplets from gathering [10]. Although there have been numerous studies on the structural and functional properties of chestnut starch, PE stabilized by SNC prepared by sulfuric acid hydrolysis of chestnut starch together with MPI has not been reported [12,13].

Therefore, the combined use of SNC and MPI for the preparation of PE may be a strategy to prevent the rapid and strong aggregation of SNC or MPI particles. Moreover, it is also a promising way to develop new edible oil-in-water PEs with SNC/MPI complex to effectively inhibit lipid peroxidation of unsaturated fatty acids in edible oils (e.g., olive oil, walnut oil, edible tea oil and macadamia oil) and deliver unstable lipophilic active substances. Quercetin, chemically known as 3,3′,4′,5,7-pentahydroxyflavone, is a naturally occurring flavonoid that is commonly found in different fruits and vegetables. The most important property of this flavonoid is its antioxidant effect. In addition, quercetin is known to have many health benefits including cancer prevention, antiallergic, anti-inflammatory, and antiviral activities [14].

In this study, the stability and interaction mechanisms of the SNC/MPI complex were firstly studied. Then, four kinds of edible oil-in-water PEs stabilized by SNC/MPI complexes were prepared and used for the encapsulation of quercetin.

## 2. Material and Methods

### 2.1. Materials

Macadamia nuts were purchased from Lincang local market (Yunnan, China). Chestnut was purchased from Tangshan Shilipai Food Co., Ltd. (Heibei, China). Olive oil was purchased from Olilyle International Trade Co., Ltd. (Beijing, China). Walnut oil was purchased from Yunlong Jinnuo Co., Ltd. (Yunnan, China). Edible tea oil was purchased from Qiyunshan Food Co., Ltd. (Jiangxi, China). Macadamia oil was purchased from Lincang Zhenkang Co., Ltd. (Yunnan, China). Quercetin (purity ≥ 98.0%) was obtained from Chengdu Munster Biotechnology Co., Ltd. (Sichuan, China). Nile Red and Nile Blue were purchased from Aladdin Industries Ltd. (Shanghai, China). MDA (malondialdehyde) content detection kit was purchased from Nanjing Jiancheng Biotechnology Co., Ltd. (Nanjing, China). All chemicals were analytical grade.

### 2.2. Preparation of Chestnut Starch

Chestnut starch was prepared according to the following steps [15]: the chestnut (700 g) was washed, peeled, and pulverized, and then 1.5 L of distilled water was added and the mixture was stirred for 1–2 min. The starch slurry was passed through a 140-mesh filter cloth and the resulting starch suspension was allowed to settle overnight at 4 °C. The supernatant was removed, and the starch granules were dispersed in 500 mL of 0.15% NaOH and precipitated at 4 °C for 2 h. The precipitated starch was resuspended in 500 mL of deionized water and the pH was adjusted to 6.5 with 1 M HCl. The starch suspension was allowed to settle, and the supernatant was discarded. The precipitated starch was washed several times with distilled water and then freeze-dried. The chestnut starch was detected by ISO 6647-1:2015 standard method to obtain the amylose and amylopectin contents were 36.3% and 51.2%, respectively [16].

### 2.3. Extraction of MPI

MPI was extracted according to the method described by Zhong et al. [17]. with minor modifications. The macadamia nut meal was passed through a 40-mesh sieve and defatted using petroleum ether under magnetically stirred conditions at room temperature. Then, 1 g of the defatted macadamia nut powder was dispersed in 25 mL of ultrapure water, and the pH was adjusted to 8.5 with 1 M NaOH. After ultrasound-assisted extraction at 35 °C for 45 min, the mixture was centrifuged at 9800× *g* for 40 min to get the supernatant. The pH of the supernatant was adjusted to 4.5 with 1 M HCl, and the precipitate was obtained after centrifugation at 8600× *g* for 40 min. The precipitate was washed with ultrapure water, then dissolved with ultrapure water, the pH was adjusted to 7.0 with 1 M NaOH and freeze-dried to get MPI powder. The protein concentration of MPI was about 21.6 g/100 g by the BCA method.

### 2.4. Preparation of Chestnut SNC by Sulfuric Acid Hydrolysis

Chestnut SNC was prepared according to the procedure described using the method of previous research [18]. Under the condition of water bath at 40 °C, 50 mL of 3.16 M H_2_SO_4_ solution and 7.35 g chestnut starch were added to a three-mouth flask and stirred at 150 r/min for 5 days. The molecular size distribution of chestnut SNC was determined by the gel permeation chromatography method (Polymer Laboratories, Inc., Amherst, MA, USA) according to a previous study [19] and the average molecular weight is 3497 Da (Figure 1).

### 2.5. Fabrication of SNC/MPI Complexes

MPI solutions were prepared by the pH cycling method [20]. First, 3.0 g MPI was dissolved in 150 mL of deionized water, and its pH was adjusted to 11.0 with 4 M NaOH. The solution was hydrated overnight and centrifuged at 4800 rpm for 10 min to get the supernatant. After that, the pH of the supernatant MPI solution was slowly lowered to 7.5 with 1 M HCl under stirring conditions and stored at 4 °C for later use. The SNC particles were slowly added to the MPI solution under continuous shearing (6000 rpm). Then, the pH of the mixture was adjusted to 3.5, 4.5, 5.5, and 6.5, respectively, and an Ultraturax T18 homogenizer (IKA, Staufen, Germany) was used to shear and disperse the mixture at 12,500 rpm for 10 min. Mixing with ultraturrax is performed at room temperature (25 ± 5 °C). The SNC/MPI complexes with different mass ratios (1:2, 1:1, 4:1, 20:1 and 40:1) were obtained and stored at 4 °C.

### 2.6. Characterization of SNC/MPI Complexes

#### 2.6.1. Contact Angle

The dispersions of SNC/MPI particles of different mass ratios were uniformly dropped onto slides and dried in an oven (GFL-230, Tianjin Lai Bo Rui Instruments Co., Ltd., Tianjin, China) at a temperature of 30 °C. A drop of water (2 μL) was injected onto the slides using a high-precision syringe. The contact angle (θ_O/W_) of the air-dried SNC/MPI complexes pellets was measured using a contact angle meter equipped with a high-speed camera (JY-PHa, Chengde Youte Instrument Co., Ltd., Chengde, China) at an ambient temperature of 25 °C and relative humidity of 50% ± 5%.

#### 2.6.2. Particle Size and Zeta (ζ) Potential

The particle size and ζ-potential of newly prepared SNC/MPI complex solutions were measured using a Zetasizer Nano-ZS90 (Malvern Instruments, Worcestershire, UK). The composite polymer solution was diluted to 1 mg/mL, and the refractive indices of the disperse phase and the continuous phase (aqueous phase) were set to 1.47 and 1.33, respectively, and the absorption index was set to 0.001 [20]. The pH of the solution was adjusted to 6.5 with 1 M HCl or NaOH before detection. All measurements were carried out at 25 °C and each value represents the mean and standard deviation (SD) of three replicate tests.

#### 2.6.3. Scanning Electron Microscopy (SEM)

The samples of each group of SNC/MPI complexes were freeze-dried, then a small amount of sample was fixed with conductive glue on the sample table, and the surface of the sample was coated with gold spray after evacuation by an ion sputtering machine. The microscopic morphology of the SNC/MPI complexes was then observed using a field emission scanning electron microscope (Nova Nano SEM 450, FEI, Hillsboro, OR, USA) at an accelerating voltage of 5.0 kV and a magnification of 500–5000.

#### 2.6.4. Fourier Transform Infrared Spectroscopy (FTIR)

The FTIR (Tensor 27, Bruker, Germany) spectroscopy test was performed by the KBr tableting methods in the wavelength range of 500–4000 cm^−1^. The SNC/MPI complex samples were vacuity dried in advance and ground into a fine powder with a mortar. Baseline corrections of the original spectra were performed using Peak fit 4.12 software. The amide I band (1700–1600 cm^−1^) was subjected to second order derivatives and Fourier deconvolution and the best Gaussian shape curve was obtained by a curve fitting procedure. The α-helix, β-fold, β-turn and random coil were assigned to the different peaks. The contents of the four secondary structure types of the SNC/MPI complexes were determined from the curve fitting results.

#### 2.6.5. X-ray Diffraction (XRD)

XRD analysis was carried out using an X-ray diffractometer (D/MAX-2200/PC, Rigaku Corporation, Tokyo, Japan) at 40 kV and 100 mA. The diffraction angle (2θ) was scanned over a range of 10° to 40° at a speed of 2°/min.

### 2.7. Preparation of SNC/MPI Pickering Emulsions

The SNC/MPI complex suspension with the best mass ratio of 20:1 was prepared as the aqueous phase and adjusted to pH 6.5. To study the effect of different types and components of oil phase on the emulsions, PEs were prepared by adding various vegetable oil fractions (10%, 30%, 50%, 70%, and 90%) into SNC/MPI suspension (6 wt%). To study the effects of SNC/MPI suspension concentrations on the emulsions, PEs were prepared by mixing 50% of oil phase with SNC/MPI complex with SNC concentrations of 6 wt%, 8 wt%, 10 wt%, 12 wt% and 14 wt%, respectively. The above PEs were obtained by homogenizing at 13,000 g for 2 min, placed in a 10 mL glass bottle, and stored at 4 °C. After 1 and 30 days of storage, the appearance and stratification of the emulsion were observed.

### 2.8. Evaluation of PEs Stabilized by SNC/MPI

#### 2.8.1. Determination of Particle Size and Zeta (ζ) Potential

Zetasizer Nano-ZS90 (Malvern Instruments, Worcestershire, UK) was used to measure the particle size and ζ-potential of the newly prepared emulsions. After the complexes were diluted to 1 mg/mL, the refractive indexes of the dispersed phase and the continuous phase were set to be 1.47 and 1.33, respectively. And the absorption index was set to be 0.001. Particle size is described by volume-weight average diameter (d4.3). Determination of ζ-potential does not require dilution. All measurements were performed at 25 °C.

#### 2.8.2. Rheological Measurements

The rheological properties of PE were characterized by the rheometer (MCR102, Anton Paar, Graz, Austria) with a conical plate geometry (diameter 50 mm, taper angle 0.04 radian, clearance 0.050 mm) at 25 °C. The shear rate varied from 0.1 s^−1^ to 100 s^−1^. The frequency scanning range is 0.1–100 rad/s, and the strain is 1.0%. The stress scanning range was 0.1–100 Pa, and the frequency was set to 1 Hz. Each sample was measured thrice, and the variation curves of storage modulus (G’) and loss modulus (G”) with stress and frequency were recorded.

#### 2.8.3. Confocal Laser Scanning Microscopy (CLSM)

The PE stabilized by SNC/MPI was stained with Nile blue and the grease was stained with Nile Red. First, the emulsion (0.5 mL) was mixed with 10 μL of Nile blue solution (dissolved in 1,2-propanediol, 1 mg/mL) and 10 μL of Nile Red solution (dissolved in 1,2-propanediol, 1 mg/mL), and then 10 μL of the stained emulsion was placed on a concave slide and covered with a glass coverslip. The CLSM (A1 Plus, Nikon, Tokyo, Japan) instrument parameters were set as follows: Nile red excitation wavelength 488 nm, Nile blue excitation wavelength 633 nm, frequency scan at a density of 1024 × 1024 and a frequency of 100 Hz.

### 2.9. Thermal Stability and Quercetin Loading

#### 2.9.1. Measurement of Peroxide Value and Secondary Oxidation Products of Oils

The peroxide value of oils was measured using the Fe (III)-thiocyanate method described by [21] with slight modifications. The corresponding oil phase was used as a control and placed in a 37 °C incubator with the PE samples to determine the oxidation products of the oils and fats over an 8-day period. First, PE (0.2 mL) and 9.8 mL of chloroform-methanol (7:3, *v*/*v*) were added to a glass tube and vortexed for 2–4 s. Then, 0.05 mL of 30% ammonium thiocyanate solution and 0.05 mL of 0.132 M ferrous chloride solution (dissolved in 10 M HCI solution) were added and vortexed for another 2–4 s. After incubation at room temperature for 5 min, the absorbance was measured at 500 nm using an ULTRAVIOLET spectrophotometer (UV-1800PC, Shanghai Mep Instrument Co., Ltd., Shanghai, China). The peroxide value of oil in the emulsion was determined using a standard curve of ferric chloride and was expressed as equivalent ferric chloride (g/mL) of peroxide per ml of oil. The secondary oxidation products in oil were measured by MDA content. Specific operations were carried out according to the instructions of the MDA content detection kit (micro method).

#### 2.9.2. Determination of Quercetin Loading

Quercetin loading was determined in accordance with the method of Jiang et al. [22] with some modifications. Briefly, 2 mL quercetin (0.1 mg/mL) was completely dissolved in olive oil, walnut oil, edible tea oil, and macadamia oil, respectively, and added to the SCN/MPI complex to prepare quercetin-loaded PEs with different oil phases. The PEs loaded with quercetin was centrifuged at 11,000× *g* for 25 min to separate free quercetin in the supernatant. The calibration curve (*y* = 0.0905 *x* + 0.0003, *R*^2^ = 0.9963) was plotted by the variation of absorbance of standard solutions of quercetin in 70% ethanol at 373 nm as a function of quercetin concentration within the range of 0–0.2 mg/mL. The loading efficiency of quercetin in PE was calculated using Equation (1):(1)$$\mathrm{Quercetin}\;\mathrm{loading}\;\mathrm{rate}\;\left(\%\right)=\frac{\mathrm{weight}\;\mathrm{of}\;\mathrm{loaded}\;\mathrm{quercetin}\;\left(\mathrm{mg}\right)}{\mathrm{weight}\;\mathrm{of}\;\mathrm{nanoparticles}\;\left(\mathrm{mg}\right)}\times 100$$

### 2.10. Statistical Analysis

All experiments were repeated three times, three parallels each, and data were expressed as mean and standard deviation (SD). Statistical analysis was carried out by SPSS 20.0. Significant differences between mean values were determined using Duncan’s test (*p* < 0.05).

## 3. Results and Discussion

### 3.1. Characterization of SNC/MPI Complexes

#### 3.1.1. Wettability

The interfacial wettability of SNC/MPI complexes was measured to evaluate its potential as a PE stabilizer. As shown in Figure 2A, the θ_O/W_ values of SNC/MPI complexes with different mass ratios were less than 90°, which was consistent with their hydrophilicity. A further increase in SNC content decreased of θ_O/W_. When the mass ratios of SNC/MPI complexes were 20:1 and 40:1, the θ_O/W_ values of SNC/MPI complexes decreased below 30°, indicating that SNC/MPI complexes with higher SNC content had greater hydrophilicity. Consistent with previous studies, θ_O/W_~90 ° was determined as the optimum value of a PE stabilizer [23]. The SNC/MPI complex with a mass ratio of 20:1 had the smallest contact angle and strongest hydrophilicity, resulting in the best interfacial adsorption and partial wettability, thus making it an effective PE stabilizer [24].

#### 3.1.2. Particle Size, Charge, Polydispersity Index (PDI), and Storage Stability Analysis

Improvements in PE stability have been reported to be more effective when surface charges are regulated by pH changes than by increasing ionic strength [25]. As shown in Figure 2D, the solutions with different SNC/MPI ratios showed similar single peaks in the particle size distribution curves, indicating a relatively concentrated distribution of particle populations (*p* < 0.05). When the pH value was 6.5, the systems with SNC/MPI ratios of 1:2, 1:1, and 4:1 exhibited flocculation and stratification after one day of storage (Figure 2B). However, the absolute value of ζ-potential the of the complex was greater than 30 (*p* < 0.05) (Figure 2C). This is mainly because both MPI and SNC were negatively charged at this pH, which contributed to the high surface potential of the emulsion. The minimum absolute value of *ζ*-potential of SNC/MPI stabilizer complex was found at SNC/MPI ratio of 40:1 (Figure 2C). The reduction in surface charge minimized electrostatic repulsion and *ζ*-potential, which can promote particle aggregation and lead to unstable emulsions [25]. The SNC/MPI stabilizer solution with a ratio of 20:1 showed no flocculation and stratification after storing for 7 days (Figure 2B). The *ζ*-potential, particle size, and PDI of the complexes with SNC/MPI ratio of 20:1 at pH 6.5 indicated that the suspension system formed under this condition may have better stability.

#### 3.1.3. FTIR

The interaction of SNC with MPI was investigated using FTIR spectroscopy. As shown in Figure 3A, MPI has a wide absorption peak at 3418 cm^−1^, which is related to the stretching vibration of N-H and O-H in the amino acid residues [26]. The absorption peaks at 2930 cm^−1^ and 2853 cm^−1^ correspond to the asymmetric stretching vibrations of saturated and unsaturated C-H bonds (including methyl, methylene, and methylene), respectively. The characteristic peaks of MPI at 1661 cm^−1^, 1529 cm^−1^, and 1401 cm^−1^ were amide I, II, and III amide vibration, indicating the characteristic structure of MPI [17]. The O-H stretching vibrations at 3418 cm^−1^ of glucose and fructose units were observed in the FTIR spectrum of SNC, and the band at 1016 cm^−1^ (C-O stretching vibration) proved the presence of the sulfonate groups. The spectral bands in the range of 3100–3500 cm^−1^ reflected the formation and change of hydrogen bonds. When MPI and SNC were combined at different mass ratios, this band shifted from 3418 cm^−1^ to 3435 cm^−1^ (SNC/MPI 1:2), 3427 cm^−1^ (SNC/MPI 1:1), 3448 cm^−1^ (SNC/MPI 4:1), 3435 cm^−1^ (SNC/MPI 20:1), and 3422 cm^−1^ (SNC/MPI 40:1), indicating the formation of new hydrogen bonds between SNC and MPI during the formation of the complexes [26]. Compared to the FTIR spectrum of MPI, the FTIR spectra of the SNC/MPI complexes had a significant change in the amide I band. The peak fitting of the amide I band of MPI and SNC/MPI complexes with different mass ratios was carried out (Figure 3C) and the secondary structure contents of the SNC/MPI complexes of different mass ratios were calculated (Table 1). These results indicated that the α-helix content of MPI showed a gradual increase with increasing SNC. Some studies have reported that the formation of protein hydrogen bonds promoted the development of the α-helix structure [27]. It can be shown that the SNC formed more hydrogen bonds with MPI at higher contents. The α-helix content of the SNC/MPI (40:1) complex increased by 9.56% compared to MPI (*p* < 0.05). There was no significant difference in the β-turn and random coil contents between SNC/MPI complexes with different mass ratios and MPI. These results suggested that the addition of SNC increased the α-helix structure of MPI by inducing a shift from β-sheet to α-helix. Generally, α-helix and β-sheet are considered “ordered” secondary structures, while β-turns and random coil are considered “disordered” protein secondary structures [28]. Results showed that the binding process of SNC and MPI involved the interconversion of “ordered” and “disordered” protein secondary structures. Alternatively, disordered β-turns and random coils are often associated with hydrophobicity on the protein surface, as these structures facilitate the exposure of hydrophobic residues [29]. The SNC/MPI (4:1) complex had a small increase in α-helix content compared with the SNC/MPI (20:1) and SNC/MPI (40:1) complexes. The early delamination of SNC/MPI (4:1) complex during the storage could be due to the increased hydrophobic interactions within the molecule.

#### 3.1.4. XRD

XRD provides information on the microstructure of the material, including the amorphous and crystalline structure. Figure 3B shows the XRD spectra of SNC, MPI, and SNC/MPI complexes with different mass ratios. SNC is a typical A-type semi-crystalline with two single broad peaks at 15° and 23° at 2θ, and double peaks at 17° and 18° at 2θ [30]. The diffraction peak with wide peak width appeared near 20.0° for pure MPI, indicating that MPI had no crystal structure. In contrast, for the SNC/MPI complexes, when the SNC/MPI mass ratio was 1:2, the peaks of 15.0°, 17.0°, and 23.0° at 2θ disappeared compared with those at SNC/MPI mass ratios of 1:1, 4:1, 20:1, and 40:1. Compared with pure SNC (the relative crystallinity was 69.21%), the peak width for SNC/MPI (1:1) near 2θ = 15.0° increased, while the peak intensity decreased. The relative crystallinity decreased to 58.37%, indicating that the SNC/MPI complexes showed an amorphous structure. The strong interaction between SNC and MPI causes the SNC/MPI complex to tend to disorder self-assembly, thus damaging the original crystal structure of SNC [28]. With the further increase in SNC content, the composites with SNC/MPI mass ratios of 4:1, 20:1, and 40:1 had strong diffraction peaks near 2θ = 15.0°, 17.0°, and 23.0°. The relative crystallinity of the composites with SNC/MPI mass ratios of 4:1, 20:1, and 40:1 gradually increased at 2θ = 23°, which were 64.03%, 65.55% and 79.34%, respectively. This is due to the increase of SNC content in the composite, resulting in larger relative crystallinity.

#### 3.1.5. Morphological Observations

The morphology of the complexes with different SNC/MPI mass ratios was observed by SEM. In the absence of SNC, the MPI granules were relatively large, irregular shapes with folded surfaces. The native chestnut starch granules were round, oval, and polygonal with smooth surfaces and had no obvious surface damage [10] Compared with regular oval onion-like native starch granules, SNC had a higher aspect ratio (the ratio of the maximum diameter to the minimum diameter) and a rough block structure (Figure 4).

SNCs with irregular and agglomerated morphology exhibited higher crystallinity and hydrophobicity [24] Therefore, SNC particles with irregular structures can stabilize the emulsion that spherical particles cannot stabilize. Starch particles with nanometer particle size and high aspect ratio shape can further improve the stability of PE [25]. As shown in Figure 4, the size of the SNC/MPI complex particles decreased first and then increased as the level of SNC increased. At the SNC to MPI mass ratio of 1:2, large lumpy binders with rough surfaces were produced. Further increases in the mass ratio of SNC to MPI to 1:1 and 20:1 resulted in irregularly shaped aggregates of the SNC/MPI complexes with a gradually decreasing and evenly distributed range of polymer diameters. At a SNC to MPI mass ratio of 40:1, the SNC/MPI complexes appeared as a large lumpy bond with a smooth surface, but rough fracture surface. This phenomenon suggests that at high SNC levels, SNC molecules cover the surface of the composite particles and cross-link together to form bulky bulk structures.

### 3.2. Preparation and Characterization of PE

#### 3.2.1. Particle Size, Charge, PDI, and Storage Stability Analysis

The average droplet size, PDI and the *ζ*-potential of PE with different oil phases stabilized by different SNC/MPI complex concentrations are summarized in Table 2. The results showed that under constant oil content, with the increase of SNC/MPI complexes concentration from 6 wt% to 14 wt%, the average droplet size and PDI of the four oil phases PE gradually decreased. The ζ-potential of PE with the four oil phases did not significantly change. This indicated that the charge of four types of oil phases is not high, and the change in the oil phase has little effect on the ζ-potential of the emulsion. The appearance of PE at different oil fractions (φ = 10%, 30%, 50%, 70%, 90%) for four different oil phases stabilized by 6 wt% SNC/MPI complexes is shown in Figure 5a.

As previously reported, PE with the higher oil fractions (φ = 70%, 90%) was unstable, showing droplets aggregating and releasing oil in the upper phase of the emulsion [31]. Apparently, the emulsified phase volume of PE stabilized by SNC/MPI complexes with four different oil phases increased with increasing oil fractions (φ from 10% to 50%). PE with olive oil and tea oil containing 50% oil phase fraction remained stable even after 30 days of storage (Figure 5a). PE with walnut oil and macadamia oil fractions (φ = 50%) both showed slight phase separation during storage for 30 days. Therefore, 50% oil fraction for the four different oil fractions of was chosen to prepare PE with long-term storage stability. In terms of the morphological structure of the appearance of fresh emulsions (Figure 5b), the close packing of four different oil phases (φ = 50%) PEs stabilized with different concentrations of SNC/MPI complexes could form a solid structure with a certain shape. The PEs of olive oil and tea oil prepared at 6 wt% to 14 wt% SNC/MPI complex concentrations and the PEs of walnut oil prepared at 10 wt% to 14 wt% SNC/MPI complex concentrations showed no emulsion delamination after 30 days. At 10 wt% SNC/MPI complex concentration, the PE of the macadamia oil phase was least precipitated.

#### 3.2.2. Rheology Measurement

Figure 6 illustrates that at frequencies between 0.1–100 rad/s, G’ was greater than G” for the different oil phase PEs, indicating that in this state the emulsions were all gel structured emulsions and had elastic dominant gel properties [32]. When the oil phase volume fraction of the four oil phase PEs was fixed, both G’ and G” increased gradually with increasing particle concentration of SNC/MPI complexes.

It is noteworthy that when the SNC/MPI complexes concentration was 14 wt%, G’ and G” of olive oil phase PE were much higher than those of the other oil phases, indicating that the dense olive oil phase emulsion structure leads to higher storage modulus and loss modulus [29]. This was attributed to the viscoelasticity of vegetable oil which related to interfacial tension, fatty acid composition, temperature, and other factors [33]. The shear thinning behavior of PE gels causes the apparent viscosity of PE in different oil phases to decrease with increasing shear rates (0.1–100 s^−1^) [34]. It can be seen that the different oil phases do not affect the shear thinning of the emulsions, which still exhibited non-Newtonian fluid behavior.

#### 3.2.3. Observation of Interfacial Structures

The different oil phases and SNC/MPI complexes were dyed with Nile red and Nile blue A, respectively. The fluorescence fields of different colors indicated the existence of the oil phase and stabilizer particles, respectively. As shown in Figure 7, the oil phase was inside the droplet, and the stabilizer particles formed a dense accumulation layer at the droplet boundary, which provided a dense barrier for oil droplets to prevent agglomeration and Ostwald ripening. Droplet coalescence was hardly detected in these PEs, which further indicated that SNC/MPI complexes were qualified PE stabilizers. Additionally, with the increase in SNC/MPI complex concentration, the average diameter size of the PE droplets with the four different oil phases gradually decreased and the distribution became more uniform and more dense.

#### 3.2.4. Thermal Stability and Quercetin Loading Analysis

For all Pes, the measured values of lipid oxidation products (PV and MDA) indicated that lipid oxidation occurred in the four oil phases during storage (Figure 8A,B). However, the oxidation of the four different oil phases of emulsion without the addition of SNC/MPI complexes occurred at a faster rate than that of PE stabilized with SNC/MPI complexes, indicating that the addition of SNC/MPI complexes inhibited lipid oxidation of PE. Pes of the olive oil and walnut oil phases had the lowest lipid primary oxidation at SNC/MPI complexes concentrations of 8 wt% and 10 wt%, respectively. While the PE of the edible tea oil had the lowest primary lipid oxidation products and the secondary oxidation products at SNC/MPI complexes concentrations of 6 wt% (Figure 8A).

The MDA content of PE in the macadamia oil phase at different SNC/MPI complex concentrations was lower than other PEs of other oil phases at various concentrations (Figure 8B). When the oil fraction is fixed, the smaller average droplet size of the emulsion may lead to a larger total interface area of the emulsion droplet [35]. The high degree lipid oxidation of PE prepared by high concentration of SNC/MPI complexes may be related to the high water activity and the accessibility of oxidant to oil droplets in the aqueous phase [36]. PEs with good storage stability and thermal stability were selected for embedding quercetin. The optimal SNC/MPI concentrations of PE with olive oil phase, walnut oil phase, edible tea oil phase, and macadamia oil phase were 8 wt%, 10 wt%, 6 wt%, and 10 wt%, respectively. Studies have reported that the encapsulation efficiency of quercetin in crude emulsion, nanoemulsion, HIPEs, and emulsion gel delivery systems is higher than 81.56% [37]. After embedding quercetin with the emulsions under the above four conditions, the loading rate was greater than 93% (Figure 8C). The embedding effect of the four emulsions was much better than that of other delivery systems.

## 4. Conclusions

This work clearly showed that SNC/MPI complex with the optimal mass ratio of 20:1 was formed by hydrophobic interactions and hydrogen bonding at pH 6.5. The addition of SNC induced the transformation of MPI from β-sheet to α-helix, and the addition of MPI α-helix structure varied with the increase of SNC. The oxidative stability of the four different edible oil phase PEs formed can be greatly improved by selecting the appropriate SNC/MPI stabilizer concentration. The optimal SNC/MPI concentrations of PE with olive oil phase, walnut oil phase, edible tea oil phase, and macadamia oil phase were 8 wt%, 10 wt%, 6 wt%, and 10 wt%, respectively. Under the optimal conditions of different edible oil PEs stabilized by SNC/MPI, they could be used as the effective carrier of quercetin, and the loading rates were all over 93%. The current work provides a new stabilizer for PE to produce good embedding delivery systems of bioactive ingredients in the functional food industry. Moreover, results obtained in this work may also provide a theoretical basis for the high value application of chestnut starch and macadamia protein isolate.

## Figures and Tables

**Figure 1 foods-11-03320-f001:**
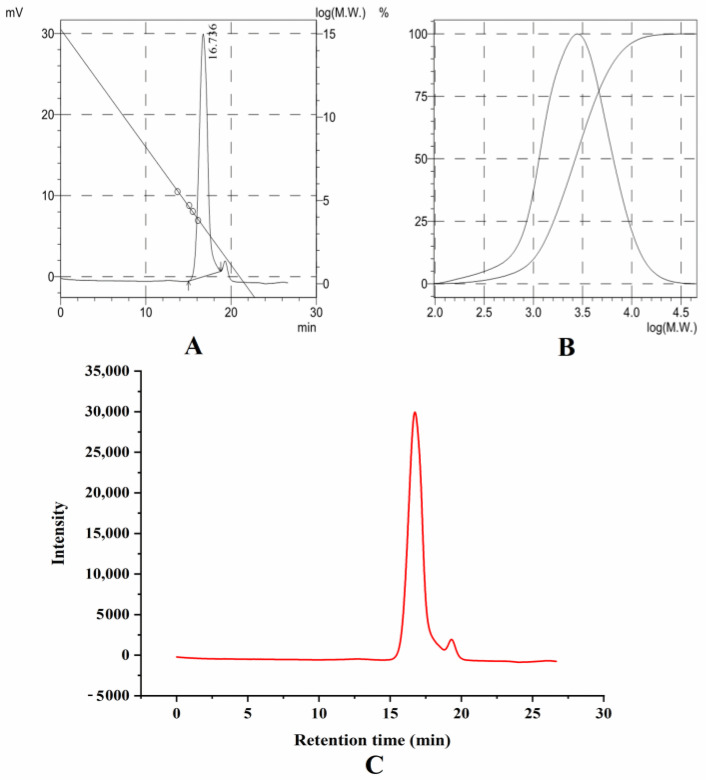
Gel permeation chromatography of chestnut starch nanocrystal. (**A**) Chromatogram and calibration curve. (**B**) Molecular weight distribution curve. (**C**) Intensity of molecular weight.

**Figure 2 foods-11-03320-f002:**
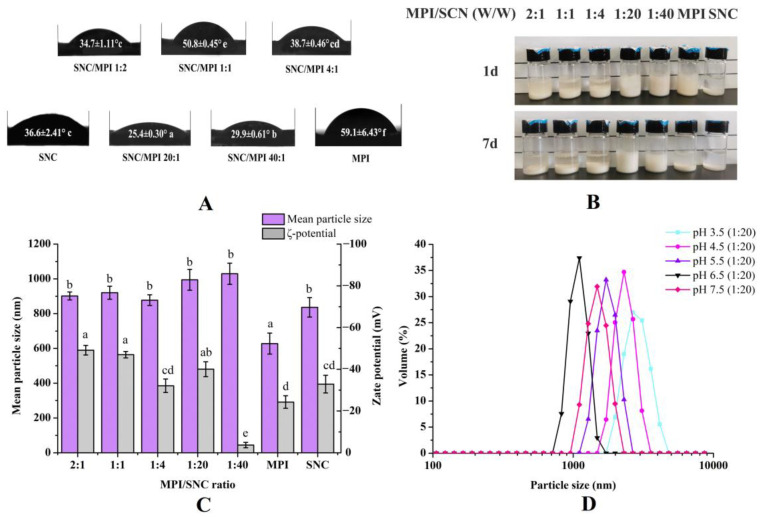
The three-phase antennae (θ_O/W_) (**A**) and storage stability (**B**) of starch nanocrystal/Macadamia protein isolates (SNC/MPI) complexes prepared using different mass ratios of SNC to MPI at pH 6.5. Mean size, Zeta (ζ)-potential (**C**) and polydispersity index (PDI) (**D**) of the SNC/MPI complexes with a ratio of 20:1 were obtained under different pH conditions. Different letters mean significant differences (*p* < 0.05).

**Figure 3 foods-11-03320-f003:**
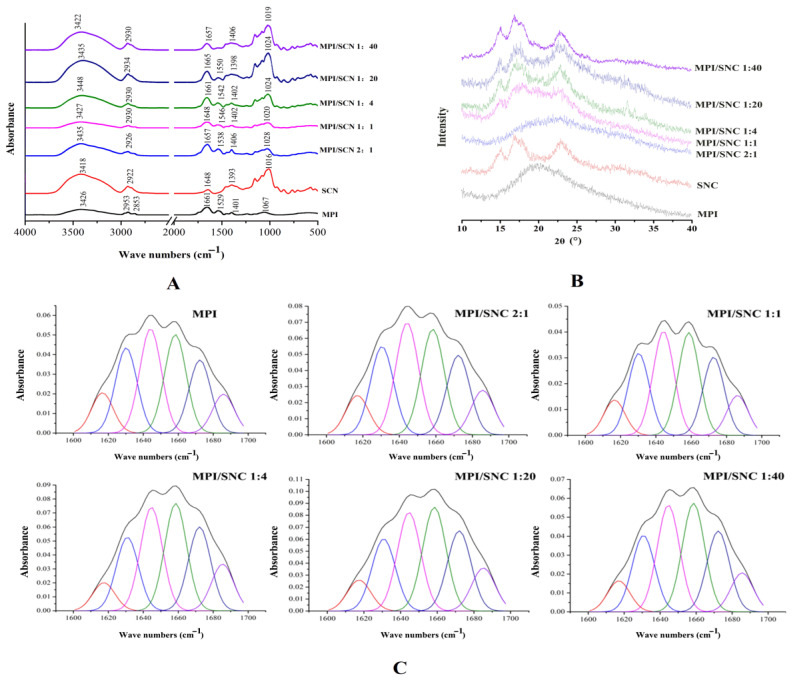
(**A**) FTIR spectra of Macadamia protein isolate (MPI), starch nanocrystal complexes (SNC) and starch nanocrystal/Macadamia protein isolates (SNC/MPI) complexes at different MPI to SNC mass ratios. (**B**) XRD diffractograms of MPI, SNC and SNC/MPI complexes at different SNC to MPI mass ratios. (**C**) Curve fitted amide I regions (1600–1700 cm^−1^) in FTIR spectra of the SNC/MPI complexes prepared at pH 6.5 with varied mass ratios.

**Figure 4 foods-11-03320-f004:**
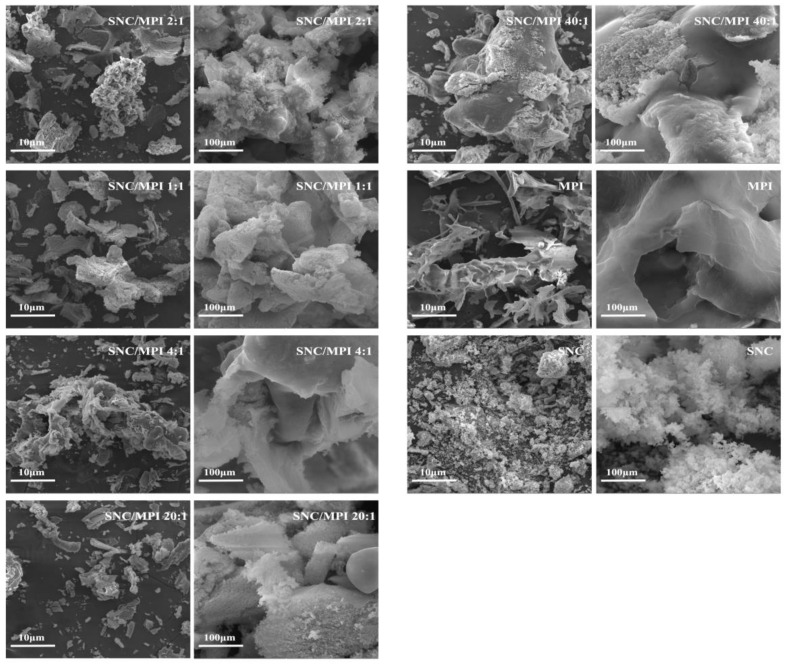
Scanning electron microscopy images of starch nanocrystal/Macadamia protein isolate (SNC/MPI) complexes prepared at pH 6.5 with varied mass ratios.

**Figure 5 foods-11-03320-f005:**
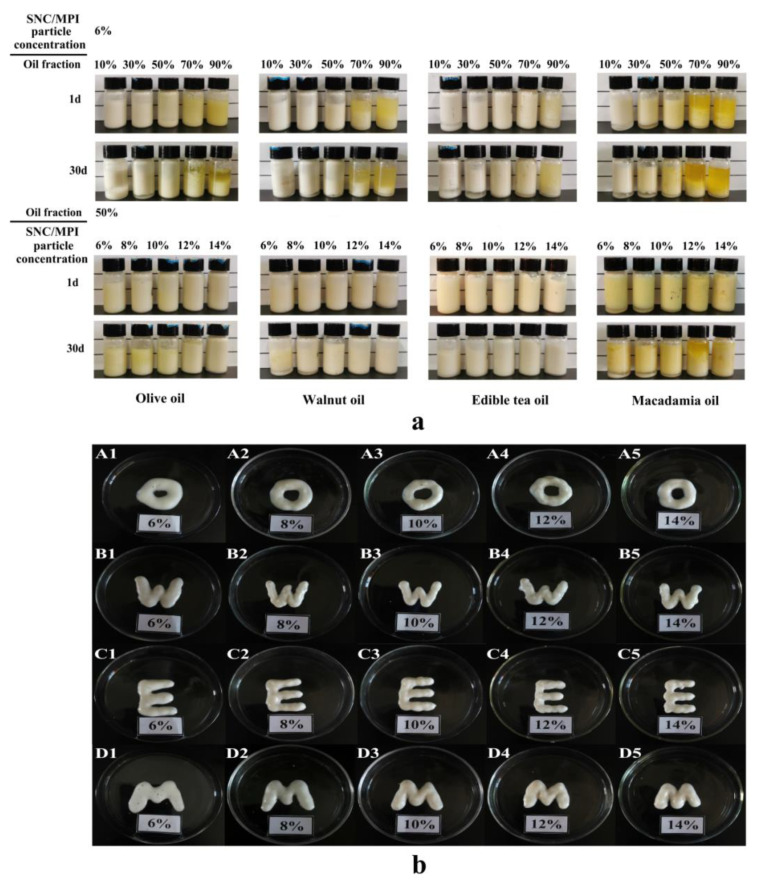
The stability of starch nanocrystal/Macadamia protein isolates (SNC/MPI) complexes stabilized Pickering emulsions (PEs) prepared with various oil phases were observed for 1 and 30 days. (**a**) Olive oil, walnut oil, edible tea oil, and macadamia oil (φ = 10–90%) with a fixed SNC/MPI complex concentration of 6 wt%. Different concentrations (6–14 wt%) of SNC/MPI complexes with fixed oil phase fractions (φ = 50%) of olive oil, walnut oil, edible tea oil, and macadamia nut oil. (**b**) Visual observations of fresh SNC/MPI complexes stabilized PEs prepared at a fixed olive oil (A1–A5), walnut oil (B1–B5), edible tea oil (C1–C5), and macadamia oil (D1–D5) fractions (φ = 50%) with different SNC/MPI complex concentrations of 6–14 wt%, respectively. Different letters mean significant differences (*p* < 0.05).

**Figure 6 foods-11-03320-f006:**
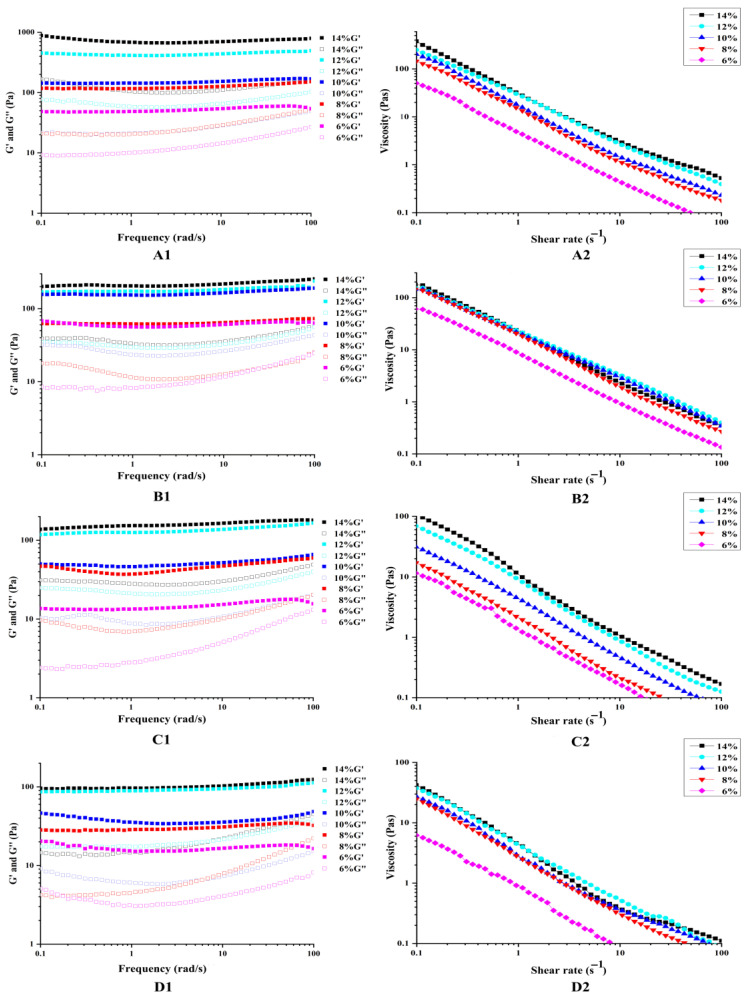
Storage moduli (G’), loss moduli (G”), and obvious viscosity of starch nanocrystal/Macadamia protein isolates (SNC/MPI) complexes stabilized Pickering emulsions of olive oil (**A****1**,**A2**), walnut oil (**B1**,**B2**), edible tea oil (**C1**,**C2**), and macadamia oil (**D1**,**D2**) at a fixed oil phase composition (φ = 50%) and different particle concentrations.

**Figure 7 foods-11-03320-f007:**
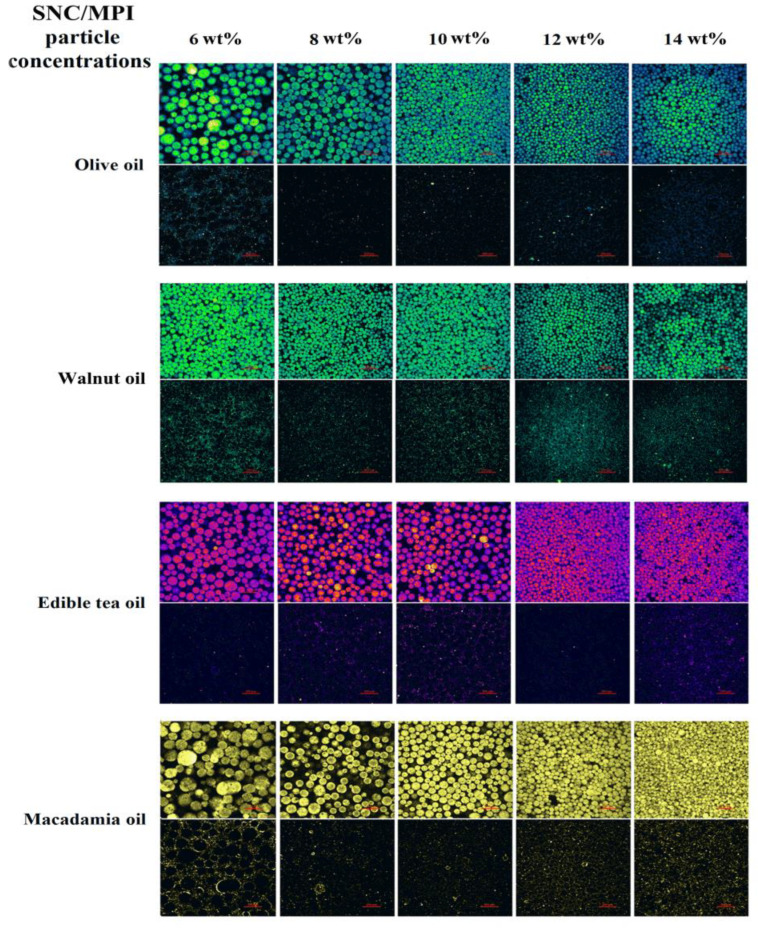
Confocal laser scanning microscope images of starch nanocrystal/Macadamia protein isolate complexes stabilizing different oil phases Pickering emulsions at different particle concentrations (6–14 wt%). O, olive oil; W, walnut oil; E, edible tea oil; M, macadamia oil.

**Figure 8 foods-11-03320-f008:**
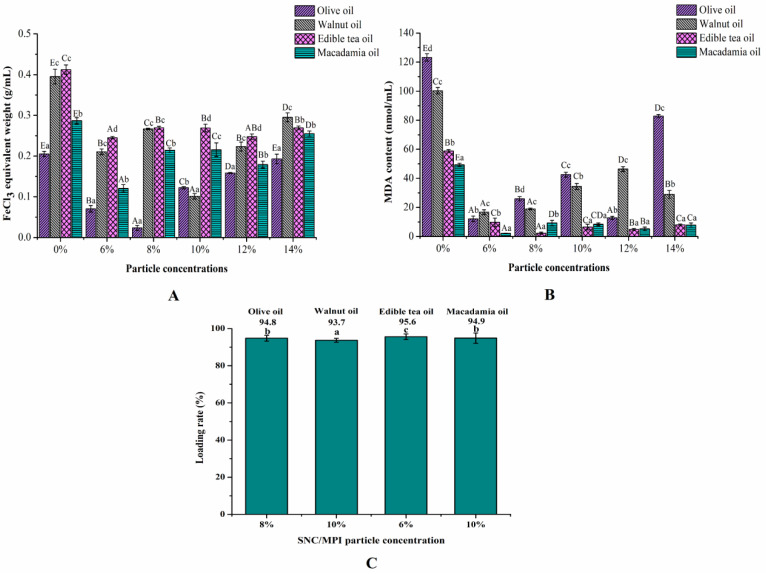
Oxidative stability of different oil phases Pickering emulsions (Pes) stabilized by starch nanocrystal/Macadamia protein isolates (SNC/MPI) complexes. The peroxide values (**A**) and thiobarbituric acid reactive substance values (**B**) of the Pes were obtained at 37 °C for 8 days, and the oil phase components were fixed at different particle concentrations (6–14 wt%). (**C**) Loading rate of Pes prepared at olive oil, walnut oil, edible tea oil, and macadamia oil fractions (φ = 50%) with different SNC/MPI complex concentrations (8 wt%; 10 wt%; 6 wt%; 10 wt%), respectively. Different letters mean significant differences (*p* < 0.05).

**Table 1 foods-11-03320-t001:** The secondary structure content of starch nanocrystal/Macadamia protein isolates (SNC/MPI) complexes with varied SNC to MPI mass ratios calculated from FTIR measurements.

**Samples Ration**	**Secondary Structure Composition (%)**			
	**α-Helix**	**β-Sheet**	**β-Turn**	**Random Coil**
MPI	22.27 ± 0.15a	28.65 ± 0.23c	25.40 ± 0.03a	23.67 ± 0.05ab
SNC/MPI 1:2	22.83 ± 0.47ab	26.96 ± 0.35bc	26.21 ± 0.31ab	24.00 ± 0.18ab
SNC/MPI 1:1	23.28 ± 0.05b	26.06 ± 0.66ab	27.02 ± 0.47ab	23.64 ± 0.14ab
SNC/MPI 4:1	23.99 ± 0.31c	23.87 ± 1.29a	28.55 ± 1.37b	23.59 ± 0.36ab
SNC/MPI 20:1	24.11 ± 0.11c	24.92 ± 1.22ab	27.60 ± 1.67ab	23.37 ± 0.56a
SNC/MPI 40:1	24.40 ± 0.16c	25.11 ± 1.13ab	26.06 ± 1.45ab	24.43 ± 0.48b

Note: values were expressed as the mean ± SD (*n* = 3); means with different letters in the same column differ significantly (*p* < 0.05).

**Table 2 foods-11-03320-t002:** Effects of different oil phases on the particle size, PDI and ζ-potential of Pickering emulsions stabilized by Macadamia protein isolates/starch nanocrystal complexes with different particle concentrations.

**Particle Concentration**		**6 wt%**	**8 wt%**	**10 wt%**	**12 wt%**	**14 wt%**
**Oil Phase**						
Olive oil	Size (nm)	2959.20 ± 118.00c	1940.33 ± 244.99b	1836.00 ± 37.36b	1678.00 ± 89.27ab	1469.10 ± 140.57a
	PDI	0.67 ± 0.15e	0.52 ± 0.22d	0.47 ± 0.15c	0.31 ± 0.02b	0.25 ± 0.11a
	ZP (mV)	30.69 ± 4.88ab	25.7 ± 3.46a	36.64 ± 1.45b	36.99 ± 1.15b	31.65 ± 2.25ab
Walnut oil	Size (nm)	2505.33 ± 260.94c	2497.10 ± 168.38c	1927.67 ± 101.57b	1542.00 ± 215.00ab	1363.50 ± 71.50a
	PDI	0.85 ± 0.24e	0.66 ± 0.29d	0.54 ± 0.03c	0.46 ± 0.14b	0.37 ± 0.04a
	ZP (mV)	28.86 ± 3.07abc	20.28 ± 3.91a	25.04 ± 5.08ab	34.78 ± 3.48c	31.16 ± 1.36bc
Edible tea oil	Size (nm)	1313.00 ± 136.00b	905.40 ± 76.60a	822.80 ± 12.04a	753.35 ± 5.85a	746.10 ± 26.00a
	PDI	0.78 ± 0.03d	0.66 ± 0.12cd	0.53 ± 0.07bc	0.40 ± 0.05ab	0.34 ± 0.01a
	ZP (mV)	20.92 ± 0.62a	25.27 ± 3.79ab	26.60 ± 0.35b	32.47 ± 2.07c	29.25 ± 0.15bc
Macadamia oil	Size (nm)	1972.50 ± 83.50c	1655.0 ± 281.00bc	1432.33 ± 206.00b	1297.00 ± 124.00b	892.05 ± 37.05a
	PDI	0.91 ± 0.01e	0.76 ± 0.07d	0.58 ± 0.09c	0.36 ± 0.07b	0.13 ± 0.03a
	ZP (mV)	23.40 ± 1.80a	23.85 ± 3.75a	23.70 ± 2.81a	24.77 ± 1.91a	25.85 ± 1.95a

Note: values were expressed as the mean ± SD (*n* = 3); means with different letters in the same row differ significantly (*p* < 0.05). PDI: polydispersity index. ZP: Zeta (*ζ*)-potential.

## Data Availability

Data is contained within the article.

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
