# Peer review of "Chestnut Starch Nanocrystal Combined with Macadamia Protein Isolate to Stabilize Pickering Emulsions with Different Oils"

_foods, 2022, doi:10.3390/foods11213320_

Round 1
Reviewer 1 Report
The article „Chestnut starch nanocrystal combined with macadamia protein isolate to stabilize Pickering emulsions with different oils“ is a reserach article focused on a preparation of o/w emulsions stabilized by Pickering particles with an additional value.
Generally, the scientific work is well described. I suggest some minor changes in the manuscript.
line 59 – I recommend that you consider adding information about quercetin (and why quercetin?) – e.g. nutritional value
line 61 – Materials – add an origin of the used quercetin
line 92 – neutral ...pH? ...I would be also useful to specify the certain value
Author Response
TO: Editor and Reviewers
Thank you for your suggestions, which are great and helpful to us. We had revised the manuscript thoroughly according to your advice, and some points which we do not change will be justified by some reasons.
Thank you very much!

Reviewer 2 Report
The results are clearly presented and interconnected in a logical manner, carefully interpreted. In my opinion this is very good scientific paper and I recommend addressing the following minor remarks.
- The author state about protein content of 21% in macadamia protein isolate , which is rather low protein content. Did the authors try to increase the value by modifying the extraction process?
- The discussion on SEM images is rather speculative since the pictures given there do not clearly support the statements. However, this analysis is not decisive for the study, so in my opinion the it can be removed.
- I recommend authors to emphasize the novelty of this work, in sense that this this blend made of starch and protein from exotic sources have never been investigated before for emulsifying properties.
Author Response

(The authors gave the same response as above.)

Reviewer 3 Report
Some considerations will be made to the authors in order to improve or clarify the article:
1 – Pickering emulsions are emulsions stabilized with solid particles. The authors report that macadamia protein isolate (MPI) is a promising natural emulsifier due to its amphiphilic properties – line 41. The main question is: if the MPI have emulsifier properties, what is the difference to the conventional emulsifiers?
2 – Line 114 – In mixture was achieved using an ultraturrax. Was temperature considered?
3 – Line 169 – Show the refractive index of the emulsion.
4 – Figure 4 – To facilitate the interpretation of figure 4, a legend in each column is recommended to understand the main difference.
5 – Line 399 - It is not perceptively that particles are at the boundary of the oil droplets. Additionally, if the authors use the same dyes to dye the emulsion, why do the Pickering droplets have different colors after acquisition?
6 – Miscellaneous – Added end point in line: 325, 327, 332.
Author Response

(The authors gave the same response as above.)
